# HAct: Out-of-Distribution Detection with Neural Net Activation Histograms

## Abstract

We propose a simple, efficient, and accurate method for detecting out-of-distribution (OOD) data for trained neural networks. We propose a novel descriptor, *HAct* - activation histograms, for OOD detection, that is, probability distributions (approximated by histograms) of output values of neural network layers under the influence of incoming data. We formulate an OOD detector based on HAct descriptors. We demonstrate that HAct is significantly more accurate than state-of-the-art in OOD detection on multiple image classification benchmarks. For instance, our approach achieves a true positive rate (TPR) of 95% with only 0.03% false-positives using Resnet-50 on standard OOD benchmarks, outperforming previous state-of-the-art by 20.67% in the false positive rate (at the same TPR of 95%). The computational efficiency and the ease of implementation makes HAct suitable for online implementation in monitoring deployed neural networks in practice at scale.

## 1 Introduction

Machine learning (ML) systems are typically constructed under the assumption that the training and test sets are sampled from the same statistical distribution. However, in practice, that is often not the case. For instance, data from new classes different from training may appear in the test set during operation. In these cases, the ML system could perform unreliably, with possible high confidence on erroneous outputs (DeVries & Taylor, 2018). It is thus desired to construct techniques so that the ML system can detect such *out-of-distribution* (OOD) data; this is called the *OOD detection problem*. Once OOD data is detected, the user of the system can be notified of potentially an unreliable result and/or other algorithms can be employed to adapt to such new data. The OOD problem has become a problem of significant recent interest in machine learning and computer vision (Yang et al., 2022), as it is important in the deployment of systems in practice.

Recent state-of-the-art (SoA) (Sun et al., 2021; Djurisic et al., 2022; Ahn et al., 2023; Sun & Li, 2022) in OOD detection for neural networks has focused on identifying descriptors of the data that can distinguish between OOD and in-distribution (ID) data. Descriptors from the data that are sufficiently different from corresponding training data descriptors are considered to be from OOD data. Because the network computes statistics of the data layer by layer to determine features that are relevant for the ML task, many recent works have hypothesized that computing functions of such statistics can be used to identify OOD data. Indeed, such approaches have led to SoA performance. One such popular approach (Sun et al., 2021) determines that a threshold of the output of activations in the penultimate layer of classification convolutional neural networks (CNNs) is an effective descriptor. This approach has been generalized to other layers (Djurisic et al., 2022) and several other recent works (Sun & Li, 2022; Ahn et al., 2023) have built upon this idea of building functions formed by thresholds of network activations. While these approaches have demonstrated SoA performance on large-scale datasets in an efficient manner that has the potential to be deployed in real-world systems, performance still needs to be advanced for use in applications such as safety-critical systems.

In this work, we introduce novel descriptors for OOD detection that are simple and efficient to compute and lead to a significant improvement in performance compared to state-of-the-art. We show that effective descriptors for OOD are probability distributions of the output values of neural network layers. We show how these descriptors can be incorporated within an efficient OOD detection

algorithm from an existing trained network. Our specific contribution is that we introduce a novel descriptor (HAct) that can be used in OOD detection, i.e., the probability distribution of the output of a layer within a neural network. When combining this descriptor over multiple layers in an OOD detection framework, the resulting technique out-performs existing state-of-the-art as demonstrated on multiple benchmark datasets.

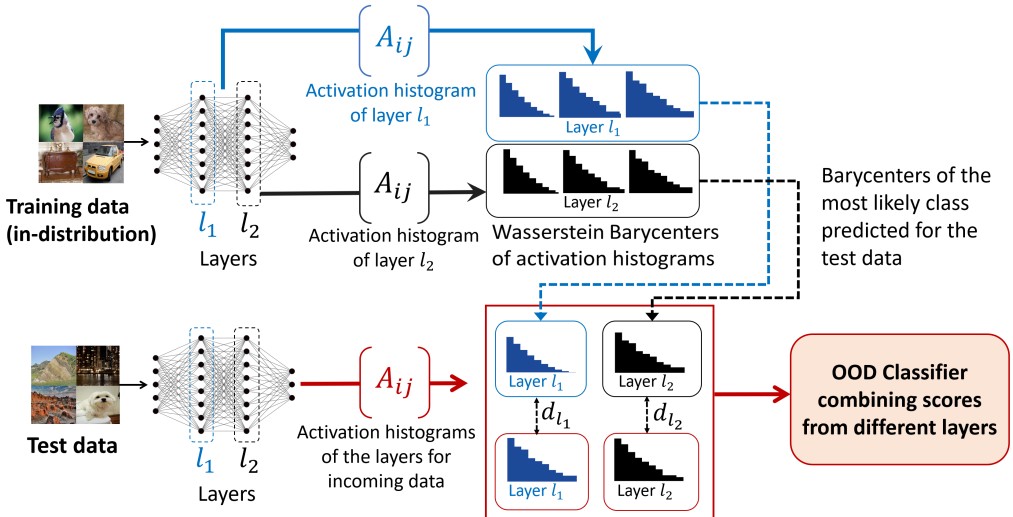

Figure 1: **Schematic of Activation Histogram (HAct) based OOD detection.** HAct involves a preparation phase in which barycenters of activation histograms for each class in the training dataset are calculated. During online operation, the HAct is computed for incoming data, and the distance to the most likely barycenter is thresholded. The illustration above shows the case where HAct descriptors are computed for two layers.

## 2    RELATED WORK

We highlight related work in OOD detection and refer the reader to (Yang et al., 2022) for a detailed survey. Several methods determine OOD data by comparing the data at test-time to the training dataset. For efficiency, this is typically implemented with auto-encoders (Zhou & Paffenroth, 2017), in which the auto-encoder is trained with ID data, which allows them to learn the distribution of the training data. Thus, any data that is OOD would have a high reconstruction error, and thus the reconstruction error is thresholded to identify OOD data. While effective, this approach may characterize data that is different from training data as OOD, yet the network might still be able to generalize to that data. Other approaches aim to model uncertainty of a network on data, and characterize high-uncertainty data as OOD. There are several methods to determine uncertainty, e.g., ensemble methods (Rahaman & Thiery, 2021) that measure the divergence of an ensemble of networks on data, test-time augmentation approaches (Wang et al., 2019) that measure the divergence of the network from augmented versions, uncertainty of network confidence scores (Malinin & Gales, 2018), and Bayesian approaches (Goan & Fookes, 2020) that treat weights in the network as probability distributions and calculate the output as the resulting distribution.

More recently, current SoA (Sun et al., 2021; Djurisic et al., 2022; Ahn et al., 2023; Sun & Li, 2022) has sought to construct what can be characterized as descriptors of the trained network and the incoming data to distinguish between OOD and ID data. In Sun et al. (2021), a threshold of activation outputs in the penultimate layer of a network is used as a descriptor for OOD detection. This is used to compute the energy score (Liu et al., 2020) (see Liu et al. (2023) for an alternative score); data with a large value of the score is labeled as OOD data. Djurisic et al. (2022) generalizes the approach of Sun et al. (2021) by thresholding not just in the penultimate layer but multiple feature layers, improving performance. Ahn et al. (2023) also thresholds activation outputs, but uses the total number of activated features as a descriptor, and uses pruning to remove un-important parts of the network (see also Sun & Li (2022) for a related idea of sparsification) for OOD and this out-performs the

results of Djurisic et al. (2022). Other descriptors of the trained network and the incoming data for OOD detection are topological descriptors (Lacombe et al., 2021), which are computed at dense layers. While effectively demonstrated on small networks, so far this approach has not scaled to large networks and datasets demonstrated in state-of-the-art (e.g., (Sun et al., 2021; Djurisic et al., 2022)). Our approach computes a descriptor as a function of the network and incoming data, but instead, we show that the distributions of outputs of layers in a network are an alternative and effective descriptor for OOD detection, improving state-of-the-art while being simple and efficient.

## 3 METHOD FOR OOD DETECTION

In this section, we present our novel approach for OOD detection. We start by presenting our novel descriptor, called *Activation Histograms* (HAct), for OOD detection, computed from the trained network and the incoming data. We then show how this descriptor can be integrated within an OOD detection procedure. Finally, we show how the computation of HAct descriptors can be made efficient for large networks. To illustrate the principles, we focus on OOD detection within the classification task.

### 3.1 ACTIVATION HISTOGRAM (HACT) DESCRIPTORS

Given a trained neural network $F$, we define a descriptor for a layer $l$ described by a linear operation within the network (e.g., linear or convolutional layers). Let $x$ be the input tensor to layer $l$, and $W$ be the weight tensor for layer $l$. We consider the layer to be formed by a linear operation as defined by $y_i = \sum_j W_{ij} x_j$ where $i, j$ represent (multi-dimensional) indices of the tensors (biases are not used in our approach). We consider the *activation weights*, formed from scalar components of intermediate computations of layer $l$, i.e., $A_{ij} := |W_{ij} \cdot x_j|$, which describes how much the $j$-th coordinate of the input observation $x$ activates the $ij$-th unit of the weight tensor. Our descriptor, called the *activation histogram*, for layer $l$ is the probability distribution of the elements of $A$, i.e., $A_{ij}$, which considers $x_j$ to be a random variable. We approximate the probability distribution of $A_{ij}$ by computing a histogram of *all* the activation weights, which we denote as $h$, defined as follows:

$$h_k = \frac{1}{n} \sum_{i,j} \mathbb{1}_{[\alpha_k, \alpha_{k+1})}(A_{ij}), \qquad (1)$$

where $n$ is the number of activation weights (elements of $A$), $0 \leq \alpha_0 < \alpha_1 < \ldots < \alpha_m$ is the partition of the range space of the weights, and $\mathbb{1}$ denotes the indicator function. The partition is fixed, except $\varepsilon := \alpha_0$, which we consider a hyper-parameter and is tuned for OOD accuracy. The vector $h = (h_k)$ is considered the OOD descriptor for layer $l$. For the sake of simplicity of notation, we do not indicate the dependence of $h$ on $l$, but that is understood. In the next section, we will make use of multiple histograms at several layers for OOD detection. For CNNs, we will use both the dense layer used for classification and convolutional layers to compute HAct descriptors.

We typically choose $\varepsilon > 0$ as in the (deep) layers we consider, the input $x$ is usually sparse (due to e.g., ReLU activations from the previous layer), which would mean the histogram is overly weighted in the first bin. We thus use a simple thresholding operation with $\varepsilon$ to decrease this influence of near-zero outputs. We study the choice in the experiments, which do not show sensitivity near optimal choices.

### 3.2 OOD DETECTION WITH ACTIVATION HISTOGRAMS

We now specify how one can use the activation histograms in the previous section to perform OOD detection. We assume a trained neural network, $F$. For simplicity, we will assume $F$ is trained for classification. Our procedure is similar to the framework of Lacombe et al. (2021), but we will use our activation histograms rather than topological descriptors to demonstrate the effectiveness of our new descriptors. The simple observation underlying our approach is that activation histograms change for OOD data compared to ID data, and therefore, our approach seeks to detect such changes in activation histograms. Figure 1 gives a schematic overview of our approach.

The procedure first consists of preparing the OOD detector using the training data set (or a subset of it) of the trained network, and then after the preparation and during online operation, the OOD

detector no-longer requires the training set. In the preparation step, an average activation histogram $\bar{h}^c$ is computed for each classification category $c$ by computing the average of activation histograms over all data in that category, i.e.,

$$\bar{h}^c = \arg\min_h \sum_{\{x_{train}:F(x_{train})=c\}} D(h, h(x_{train})), \qquad (2)$$

where $h(x_{train})$ is the activation histogram of $x_{train}$, and $D$ is a metric between probability distributions. During online operation for OOD detection, the most likely category $c^*$ for the test data $x_{test}$ is chosen, the activation histogram for $x_{test}$, $h(x_{test})$, is computed, and if the distance between $\bar{h}^{c^*}$ and $h(x_{test})$ exceeds a threshold, the data $x_{test}$ is considered OOD for the network $F$. More formally, our OOD detection using activation histograms for a given layer $l$ is given by

$$d_l(x) = \begin{cases} \text{OOD} & D(\bar{h}^{F(x)}, h(x)) > \tau \\ \text{ID} & D(\bar{h}^{F(x)}, h(x)) \le \tau \end{cases}, \qquad (3)$$

where $\tau > 0$ is a threshold. To combine information from multiple layers for an OOD detector, we define the overall detector to return ID if all the OOD detectors for all layers return ID, otherwise, the detector returns OOD.

In the above formulation, one needs to choose an appropriate metric $D$ to define the average descriptors for training categories, and the distance between test and training histograms. Following Lacombe et al. (2021), we choose the Wasserstein metric (entropy regularized for fast computation; Benamou et al. (2015) is used for barycenter computation and Bonneel et al. (2011) for distance computation during inference), which was shown to be effective. In this case, $\bar{h}^c$ are referred to as Wasserstein barycenters. Algorithm 1 shows the pseudo-code of our proposed OOD detector inference and preparation.

---

**Algorithm 1** Pseudo-Code for HAct OOD detection

---

1: *Inputs* : Training dataset $(X, y)$, where $X$ denotes inputs, and $y \in \{1, 2, \cdots, k_0\}$ are class label outputs. Neural network $F$ trained on $(X, y)$.
2: *Preparation step*:
3: **for** layers $l_1, l_2, \cdots l_n$ **do**
4:     **for** $k = 1$ to $k_0$ **do**
5:         Calculate the Wasserstein barycenter $\bar{h}^c$ equation 2 from $(X, y)$ and $F$
6:     **end for**
7: **end for**
8: *Online Operation*: For a test observation $x$
9: **for** layers $l_1, l_2, \cdots l_n$ **do**
10:     Calculate HAct, $h(x)$ using equation 1 for layer
11:     Calculate the detector $d_l(x)$ using equation 3
12: **end for**
13: Define an overall OOD detector $d(x)$ by combining $d_l(x)$ over layers $l_1, \ldots, l_n$:

$$d(x) = \begin{cases} \text{ID} & d_l(x) = \text{ID} \ \forall l \in \{l_1, \ldots, l_n\} \\ \text{OOD} & \text{otherwise} \end{cases} \qquad (4)$$

---

### 3.3 SPEEDING-UP HAct: SUB-SAMPLING ACTIVATION WEIGHTS

For large networks, e.g., used for image classification, activation histograms computed from convolutional layers can be computationally expensive, both for training and inference. This is because the output shape of convolutional layers can typically result in activation weight matrices of size $\mathcal{O}(10^6) - \mathcal{O}(10^{10})$. To reduce the cost of computation, we propose a sub-sampling of activation weights used to compute activation histograms. This can significantly accelerate the training and test speeds when activation histograms are calculated on the convolutional layers. In the ablation studies (Section 4.2), we show the trade-off between computation cost and accuracy of this strategy.

We sub-sample activation weights by skipping some of the input-output connections $A_{ij}$ of a layer based on a predefined strategy. The strategy involves sub-sampling both input and output nodes in the computation of the activation weight matrix $A_{ij}$. We can perform sub-sampling in either of the

height ($h$), width ($w$), and channel ($c$) dimensions of a convolutional layer input/output. However, in practice, most of the convolutional layers that are used to form HAct descriptors in our experiments are close to the classification (dense) layer, and thus have small spatial dimensions (e.g., $7 \times 7$ for the last convolutional layer of ResNet-50), as compared with their channel dimensions (2048 for the last convolutional layer of ResNet-50). Therefore, we sub-sample the activation weights by sub-sampling the channels in the input and output nodes of convolutional layers and using only the weights $A_{ij}$ formed from sub-sampled $i$ and $j$ as illustrated in Figure 2. Note that in the implementation, we rasterize the input and output nodes. The connections between the rasterized input and output illustrate the activation weights $A_{ij}$ that are computed. After sub-sampling in the channel dimension, note that fewer weights $A_{ij}$ are computed and thus fewer weights are binned in histogram computation.

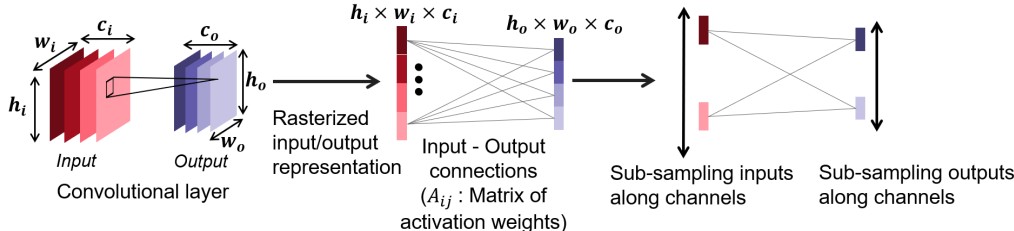

Figure 2: Schematic of the sub-sampling procedure for speeding up HAct computation on convolutional layers. The input and output nodes of a convolutional layer are sub-sampled in the channel dimension, and the activation weights are computed from the resulting sub-sampled nodes.

## 4 EXPERIMENTS

We start by comparing our method to state-of-the-art methods for OOD detection on public benchmarks (Section 4.1) and then perform a detailed ablation of our method's hyper-parameter and design choices (Section 4.2).

### 4.1 COMPARISON TO STATE OF THE ART

#### 4.1.1 DATASETS

We test our method on benchmark datasets for OOD detection. The first benchmark is derived from CIFAR-10 (Krizhevsky). Networks are trained on 8 categories of CIFAR-10 training set (denoted CIFAR-8) and the remaining two categories are considered OOD (denoted by CIFAR-2). The test set used is the CIFAR-10 test set. The second set of datasets (used in Sun et al. (2021)) involves ImageNet (Deng et al., 2009) for ID and the Places (Zhou et al., 2018), SUN (Xiao et al., 2010), iNaturalist (Van Horn et al., 2018) and Textures (Cimpoi et al., 2014) for OOD data.

#### 4.1.2 METRICS

We benchmark HAct using standard metrics for detection used in SoA (Sun et al., 2021). The first is false positive rate (FPR) at a true positive rate (TPR) of 95%, denoted as FPR95 (lower is better), and the second is the area under the ROC (receiver operating characteristic) curve, denoted as AUROC (higher is better).

#### 4.1.3 COMPARISON TO SoA ON LARGE-SCALE OOD DATASETS

We experiment with ResNet-50 (He et al., 2016) and a comparatively lighter weight model MobileNet-v2 (Sandler et al., 2018) trained on ImageNet-1k data, and benchmark on the OOD datasets used in Sun et al. (2021). We compare to the most recent SoA - ReAct (Sun et al., 2021), ASH-B (Djurisic et al., 2022), DICE (Sun & Li, 2022), DICE+ReAct(Sun & Li, 2022) and LINe (Ahn et al., 2023). For ResNet-50, HAct descriptors are combined (as specified in Algorithm 1 in equation 4) from the last classification layer which is a dense layer and the convolutional layer

preceding the dense layer. For MobileNet-v2, we combine the HAct descriptors from the last classification layer (dense layer), and the two preceding convolutional layers.

Table 1 shows the results of the benchmark comparison. Our approach consistently outperforms all competing methods by a wide margin, nearly achieving zero FPR95 and nearly 100% AUROC. Our result is a 20.67% improvement in FPR95 for ResNet-50 and 29.52% for MobileNet-v2 over the next best method (Ahn et al., 2023), demonstrating the utility of our activation histograms (HAct) descriptors.

Our approach runs in 90ms on ResNet-50 and 60ms on MobileNet-v2 for inference on a single image of size 224×224×3 on an NVIDIA GeForce RTX 3080 (see Section 4.2.3 for details), which is comparable to existing state-of-the-art.

Table 1: Results of OOD detection using ResNet-50 and MobileNet-v2 trained on Imagenet-1k as ID and various OOD datasets indicated. HAct consistently out-performs existing methods across datasets and architectures.

| | OOD Datasets | | | | | | | | | |
| | iNaturalist | | SUN | | Places | | Textures | | Average | |
| Methods | FPR95 ↓ | AUROC ↑ | FPR95 ↓ | AUROC ↑ | FPR95 ↓ | AUROC ↑ | FPR95 ↓ | AUROC ↑ | FPR95 ↓ | AUROC ↑ |
|---|---|---|---|---|---|---|---|---|---|---|
| **Model: ResNet-50** | | | | | | | | | | |
| ReAct Sun et al. (2021) | 20.38 | 96.22 | 24.20 | 94.20 | 33.85 | 91.58 | 47.30 | 89.80 | 31.43 | 92.95 |
| ASH-B Djurisic et al. (2022) | 14.21 | 97.32 | 22.08 | 95.10 | 33.45 | 92.31 | 21.17 | 95.50 | 22.73 | 95.06 |
| DICE Sun & Li (2022) | 25.63 | 94.49 | 35.15 | 90.83 | 46.49 | 87.48 | 31.72 | 90.30 | 34.75 | 90.77 |
| DICE + ReAct Sun & Li (2022) | 18.64 | 96.24 | 25.45 | 93.94 | 36.86 | 90.67 | 28.07 | 92.74 | 27.25 | 93.40 |
| LINe Ahn et al. (2023) | 12.26 | 97.56 | 19.48 | 95.26 | 28.52 | 92.85 | 22.54 | 94.44 | 20.70 | 95.03 |
| **HAct (Ours)** | **0.02** | **99.99** | **0.02** | **99.99** | **0.02** | **99.99** | **0.07** | **99.95** | **0.03** | **99.98** |
| **Model: MobileNet-v2** | | | | | | | | | | |
| ReAct Sun et al. (2021) | 42.40 | 91.53 | 47.69 | 88.16 | 51.56 | 86.64 | 38.42 | 91.53 | 45.02 | 89.47 |
| ASH-B Djurisic et al. (2022) | 31.46 | 94.28 | 38.45 | 91.61 | 51.80 | 87.56 | 20.92 | 95.07 | 35.66 | 92.13 |
| DICE Sun & Li (2022) | 43.09 | 90.83 | 38.69 | 90.46 | 53.11 | 85.81 | 32.80 | 91.30 | 41.92 | 89.60 |
| DICE + ReAct Sun & Li (2022) | 32.30 | 93.57 | 31.22 | 92.86 | 46.78 | 88.02 | 16.28 | 96.25 | 31.64 | 92.68 |
| LINe Ahn et al. (2023) | 24.95 | 95.53 | 33.19 | 92.94 | 47.95 | 88.98 | 12.30 | 97.05 | 29.60 | 93.62 |
| **HAct (Ours)** | **0.12** | **99.97** | **0.07** | **99.98** | **0.12** | **99.97** | **0.01** | **99.99** | **0.08** | **99.98** |

### 4.1.4 COMPARISON WITH TOPOLOGICAL DESCRIPTORS FOR OOD DETECTION

We compare our approach to OOD detection with topological descriptors (Lacombe et al., 2021). Note that this approach has not been demonstrated on large-scale OOD benchmarks (Sun et al., 2021) for classification, because of the large computational complexity. Since we use a similar framework, but different descriptors, we compare against (Lacombe et al., 2021) to illustrate the scalability of our approach to large-scale datasets and the advantage over that approach even for smaller-scale datasets/architectures. The topological approach can only be applied to dense layers in large CNNs due to speed limitations. Even in this case, HAct is ∼50 times faster on ResNet-50 trained on ImageNet-1k on the last dense layer. Due to the nonlinear increase of computational complexity of Lacombe et al. (2021) as a function of the number of activation weights (see Appendix), it does not scale to convolutional layers. Besides the advantage of speed, our approach is also more accurate in OOD detection. To demonstrate this, we compare against Lacombe et al. (2021) on smaller datasets (FMNIST/MNIST benchmark, where ID data is from MNIST (LeCun, 1998) and OOD is from FMNIST (Xiao et al., 2017), and the CIFAR-8/CIFAR-2 benchmark) and architectures (CNN-1 (by PyTorch.org) and CNN-2 (by TensorFlow.org) which are shallow CNNs - see Appendix for full specifications) considered in Lacombe et al. (2021). The descriptors for HAct are only computed on the final dense layer for fair comparison. Results are shown in Table 2, and indicate that our method significantly outperforms the topological descriptors (Lacombe et al., 2021).

Table 2: Comparison of our HAct descriptors with topological descriptors (Lacombe et al., 2021) for OOD detection from dense layers on the CIFAR-8/CIFAR-2 OOD benchmark.

| | | | Metric | |
| Model | OOD | Method | FPR95 ↓ | AUROC ↑ |
|---|---|---|---|---|
| CNN-1 | FMNIST | Topology | 46.50 | 92.20 |
| | | HAct (Ours) | **12.50** | **95.50** |
| CNN-2 | CIFAR-2 | Topology | 86.00 | 55.39 |
| | | HAct (Ours) | **82.25** | **67.12** |

## 4.2 ABLATION STUDIES

In this section, we thoroughly examine hyper-parameter and design choices in our method.

### 4.2.1 LAYER CHOICE FOR ACTIVATION HISTOGRAMS

As discussed in Section 3, HAct descriptors can be computed at any linear layer. We explore this choice by experimenting with the last dense layer, the convolutional layers just before the dense layer, and the combination of both using the combined detector as described in Section 3.

We start by experimenting with an OOD benchmark dataset in Table 1 (in-distribution data being ImageNet-1k and OOD being iNaturalist) with ResNet-50 by combining the dense layer with the last convolutional layer (denoted as Dense + Conv-1), and also by combining the dense layer with the convolutional layer before the last convolutional layer (denoted as Dense + Conv-2). We report the results in Figure 3(a) for OOD detection with the iNaturalist dataset. We see that the combined detector using either Conv-1 or Conv-2 layer with the dense layer results in similar OOD detection performance (with the ROC curves overlapping for the combined detectors), and that combination out-performs HAct computed only on any one layer.

For OOD detection with MobileNet-v2 (see Figure 3(b)), the best performance with HAct descriptors was from the last classification (dense) layer combined with the descriptors from the preceding two convolutional layers. This might be explained by the fact that in MobileNet-v2, separable convolutions are used and each convolutional layer is a separable part of a full convolution. Using both convolutional layers may have a similar effect as computing HAct on a full convolution layer as done in ResNet-50. Note the combination of dense with either convolutional layer has slightly inferior performance than using all three layers, and both outperform HAct computed on any one layer.

Next, we show that this result of the combination of dense and convolutional HAct descriptors improving on either individual layer HAct descriptor is not restricted to ImageNet-1k trained networks. To show this, we experiment on the CIFAR OOD benchmark with the CNN-2 architecture (mentioned in the previous sub-section). Results are shown in Figure 3(c) and indicate that the combination of HAct from both layers leads to better OOD detection performance.

The number of layers required for defining our combined detector based on HAct descriptors can be dependent on the architecture as we have seen with ResNet-50 and MobileNet-v2.

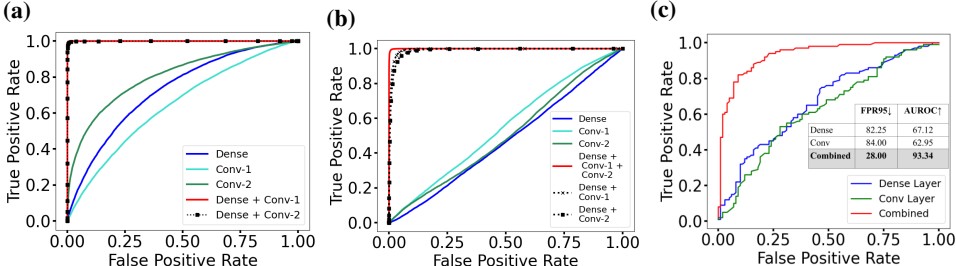

Figure 3: ROC curves using HAct descriptors over multiple layers improves OOD detection. Experiment on (a) ResNet-50 on iNaturalist OOD dataset (b) MobileNet-v2 on iNaturalist OOD dataset, and (c) CIFAR OOD dataset and a shallow CNN (CNN-2 : see text for details).

### 4.2.2 HYPER-PARAMETER SELECTION

The main hyper-parameter that needs to be tuned in our algorithm is $\varepsilon$, the threshold for near-zero values in the computation of activation histograms. In general, the optimal value of $\varepsilon$ will depend upon the architecture and layer within the architecture as the expected distribution of activation weights will vary based on these aforementioned variables. Therefore, we study the selection of $\varepsilon$ for various layers and architectures.

We study $\varepsilon \in \{0, 10^{-4}, 10^{-3}, 10^{-2}, 10^{-1}\}$ on ResNet-50 and MobileNet-v2 trained on Imagenet-1k, and report the variation on the OOD benchmark datasets used by Sun et al. (2021). We report the average AUROC and FPR95 metrics over the OOD datasets. The thresholds for both the dense and convolutional layers are varied. Results are summarized in Table 3, where $\varepsilon$ was varied for the

last classification (dense) layer and the convolutional layer before it (Conv-1). In the first half of the table, we fix $\varepsilon$ for the Conv-1 layer and vary $\varepsilon$ with respect to the dense layer, and vice-versa for the second half.

From the tables, we can see that the optimal $\varepsilon$ is not 0 for all layers/architectures, illustrating the need to address the issue of a large mass in the histograms at zero due to ReLu activations. The tables also show that the results do not change much around small variations of the optimal threshold. The optimal threshold shows dependence on layer and architecture, as expected. It appears that ResNet-50 is less sensitive to changes in $\varepsilon$ than MobileNet-v2; at this time, we do not have an explanation for this observation. We used these optimal thresholds in our benchmark comparison to SoA in the previous sub-section. Specifically, we chose $\varepsilon = 10^{-4}$ for the dense and $10^{-3}$ for the convolutional layer in ResNet-50 and $10^{-4}$ for the dense layer and $10^{-1}$ for the convolutional layers of MobileNet-v2.

Note that for ResNet-50, the maximum values of a significant proportion of the activation weights obtained from the Conv-1 layer were less than $10^{-1}$, and thus it is not a meaningful threshold, which is why it is left blank in Table 3.

Table 3: Comparison of different $\varepsilon$ values on the dense layer of ResNet-50 and MobileNet-v2 for OOD detection, averaged over the OOD benchmark datasets.

| $\varepsilon$ | ResNet-50 $\varepsilon$ variation on dense layer | | MobileNet-v2 $\varepsilon$ variation on dense layer | | ResNet-50 $\varepsilon$ variation on Conv-1 layer | | MobileNet-v2 $\varepsilon$ variation on Conv-1 layer | |
|---|---|---|---|---|---|---|---|---|
| | FPR95 ↓ | AUROC ↑ | FPR95 ↓ | AUROC ↑ | FPR95 ↓ | AUROC ↑ | FPR95 ↓ | AUROC ↑ |
| 0 | 0.05 | 99.98 | 0.15 | 99.95 | 23.26 | 95.29 | 0.52 | 99.88 |
| $10^{-4}$ | **0.03** | **99.98** | **0.08** | **99.98** | 0.03 | 99.98 | 0.42 | 99.89 |
| $10^{-3}$ | 0.05 | 99.97 | 0.10 | 99.97 | **0.03** | **99.98** | 0.61 | 99.85 |
| $10^{-2}$ | 0.22 | 99.91 | 0.20 | 99.94 | 0.03 | 99.98 | 0.24 | 99.92 |
| $10^{-1}$ | 1.39 | 99.65 | 10.29 | 98.05 | - | - | **0.08** | **99.98** |

### 4.2.3 SELECTION OF SUB-SAMPLING AMOUNT FOR ACTIVATION WEIGHTS

We perform an ablation to study the choice of sub-sampling amount in activation histograms of convolutional layers. As mentioned earlier, we sub-sample the channel dimension of input/output nodes. The primary motivation for sub-sampling as discussed earlier is computational speed.

We study the sub-sampling for three CNN architectures (a small CNN, CNN-2, specified earlier, ResNet-50 and MobileNet-v2). CNN-2 is trained on the CIFAR-8 dataset and tested on CIFAR-2 and other two are trained on ImageNet-1k, and the average over the OOD benchmarks in Sun et al. (2021). We study the trade-off between speed and accuracy, as accuracy is expected to decrease with fewer activation weights. The experiments have been performed on an NVIDIA GeForce RTX 3080 GPU, and the inference times are reported (for a single $224 \times 224 \times 3$ image for ResNet-50 and MobileNet-v2, and on a $32 \times 32 \times 3$ image for the CNN-2 architecture). Results are reported in Table 4.

The first result in Table 4 on CNN-2 shows that the cost decreases by about 2x for each sub-sampling factor of 2, as expected. We uniformly sub-sample the last convolutional layer with respect to channel input/output nodes before the dense/classification layer. Interestingly, the accuracy of OOD detection does not uniformly go down with increasing sub-sampling rate, and the optimal accuracy is not obtained with no sub-sampling (sub-sampling rate of 1 in the table). After a large amount of sub-sampling, the results degrade as expected. The user may choose a sub-sampling rate based on the computational budget allocated.

The next results in Table 4 show the trade-off in OOD detection performance with respect to the computational cost by sub-sampling the last convolutional layer (Conv-1) of ResNet-50 and MobileNet-v2. We uniformly sub-sample the channels of the last convolutional layer with a sub-sampling rate of $\{16, 32, 64\}$; due to computational cost of the experiments, we were unable to run on sub-sampling rates below 16. We found the best OOD detection performance (in terms of FPR95 and AUROC) for both the models with a sub-sampling rate of 32, consistent with the previous experiment that lower sampling rate does not necessarily yield better accuracy. Again the computational cost reduces by a factor of 2 for each sub-sampling by a factor of 2. The OOD detection accuracy does not vary much

Table 4: Comparison of different sub-sampling rates on the last convolutional layer (Conv-1) of CNN-2 trained on CIFAR-8/test on CIFAR-2, and ResNet-50 and MobileNet-v2, trained with ImageNet-1k as in-distribution. The performance metrics are shown after combining the HAct descriptors for both the models as discussed in Section 4.2.1.

| Model | Sub-sampling Rate | Metric (Combined) | | Computational cost |
|---|---|---|---|---|
| | | FPR95 ↓ | AUROC ↑ | Inference time (ms) ↓ |
| CNN-2 | 1 | 28.00 | 93.34 | 92 |
| | 2 | 24.00 | 94.42 | 45 |
| | 4 | 29.00 | 92.98 | 23 |
| | 8 | 37.01 | 93.00 | 12 |
| | 16 | 42.00 | 90.94 | 7 |
| ResNet-50 | 16 | 0.11 | 99.95 | 182 |
| | 32 | **0.03** | **99.98** | 90 |
| | 64 | 0.05 | 99.98 | 48 |
| MobileNet-v2 | 16 | 0.09 | 99.97 | 112 |
| | 32 | **0.08** | **99.98** | 60 |
| | 64 | 0.11 | 99.97 | 35 |

between 32 and 64 sub-sampling rates, perhaps indicating extraneous information for the purpose of OOD-detection.

Based on these results, we sub-sample the Conv-1 layer by 32 for both ResNet-50 and MobileNet-v2 for the benchmark with SoA in the previous section. We also chose a sub-sampling rate of 32 for the Conv-2 layer of MobileNet-v2.

## 5 CONCLUSIONS

We introduced a new descriptor HAct (activation histograms of linear layer outputs) of incoming data to a neural network that are effective in distinguishing OOD from ID data. The combination of descriptors from the dense and the preceding convolutional layer in CNNs was shown effective for OOD detection, out-performing SoA. The simplicity and efficiency of HAct imply the potential to be deployed in practical systems. Given the generality, future work will involve application to other classes of neural nets.

A current limitation of our method is the need for accessing training data in the preparation step before inference, which may preclude some applications. However, there may be methods to approximate barycenters from the trained network without the need for accessing training data, which is a subject of future work.

After review, we will make our code publicly available.

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

## A  APPENDIX

### A.1  NETWORK ARCHITECTURES

Table 5 shows the architectures of CNN-1 and CNN-2 used in our work. We follow the baseline architecture demonstrated at https://github. com/pytorch/examples/tree/master/mnist, and modify the number of neurons in the last layer for the ablation study on the effect of the size of activation matrix (Section A.3). For CNN-2, we follow the baseline architecture demonstrated at https://www.tensorflow.org/tutorials/images/cnn, and modify the last classification layer to have 8 neurons, which is suitable for our in-distribution training set of 8 classes from the CIFAR-10 image dataset (denoted as CIFAR-8). The OOD classes we chose for creating the OOD dataset (denoted as CIFAR-2) from CIFAR-10 are "*deer*" and "*ship*".

Table 5: Model architectures of CNN-1 and CNN-2

| Model | Layer | Kernel Size | Filters | Neurons | Activation |
|---|---|---|---|---|---|
| CNN-1 | Convolution 1 | $3 \times 3$ | 32 | - | Relu |
|  | $2 \times 2$ Max-Pool | - | - | - | - |
|  | Convolution 2 | $3 \times 3$ | 64 | - | Relu |
|  | $2 \times 2$ Max-Pool | - | - | - | - |
|  | Dropout (0.25) | - | - | - | - |
|  | Flatten | - | - | - | - |
|  | Fully Connected | - | - | 100 | Relu |
|  | Fully Connected | - | - | 10 | Softmax |
| CNN-2 | Convolution 1 | $3 \times 3$ | 32 | - | Relu |
|  | $2 \times 2$ Max-Pool | - | - | - | - |
|  | Convolution 2 | $3 \times 3$ | 64 | - | Relu |
|  | $2 \times 2$ Max-Pool | - | - | - | - |
|  | Convolution 2 | $3 \times 3$ | 64 | - | Relu |
|  | Flatten | - | - | - | - |
|  | Fully Connected | - | - | 64 | Relu |
|  | Fully Connected | - | - | 8 | Softmax |

### A.2  EFFECT OF THE SIZE OF ACTIVATION MATRIX : COMPUTATIONAL COST AND OOD DETECTION ACCURACY

Computational cost is a key metric to monitor for deploying an OOD detector in practice. In this section, we investigate how the OOD detection accuracy and the computational cost of our proposed

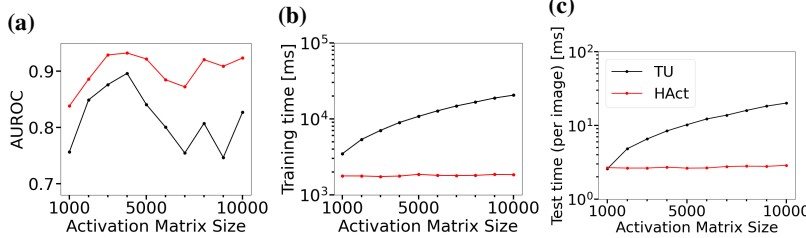

Figure 4: (a) Comparison of AUROC : HAct vs TU. (b) Training time comparison (c) Test time comparison (per image)

approach behave as a function of the size of the activation matrix, and compare the results with topological uncertainty (TU) of Lacombe et al. Figure 4 shows the results for which our approach (HAct) has been compared with TU on the last dense layer of DNN-1 for in-distribution MNIST and OOD FMNIST datasets. The size of the activation matrix is controlled by the number of neurons in the penultimate layer of CNN-1, which has been varied from 100 to 1000 in steps of 100, which resulted in the activation matrix sizes of 1000 to 10000 (the last layer of CNN-1 has 10 neurons). Each of these variations yields a new classifier network architecture, which has been trained on the MNIST dataset for 100 epochs using Adam optimizer. It is seen that HAct results in better OOD detection as evinced by higher AUROC, with significant savings in the training and test time requirements. Moreover, Figure 4 (b) and (c) indicate that the computational time requirements for training and testing using TU have a nonlinear rate of increase with the activation matrix size, while HAct has a significantly lower computational cost, with the maximum cost savings up to 10x for training and 7x for testing. This shows that HAct is a significantly faster approach than TU for the OOD generalizability of trained networks, along with potentially improved OOD detection performance.

## A.3 EFFECT OF VARYING $\varepsilon$ ON THE CONVOLUTIONAL LAYERS OF MOBILENET-V2.

Table 6: Comparison of the effect different $\varepsilon$ values chosen for Conv-1 and Conv-2 layers of MobileNet-v2 for OOD detection. FPR95 and AUROC metrics averaged over the OOD benchmark datasets (Sun et al., 2021). Conv-2 layer sub-sampled with a rate of 16 (top) and 32 (bottom).

| | FPR95↓ | | | | | | AUROC↑ | | | | |
|---|---|---|---|---|---|---|---|---|---|---|---|
| $\varepsilon$ | 0 | $10^{-4}$ | $10^{-3}$ | $10^{-2}$ | $10^{-1}$ | $\varepsilon$ | 0 | $10^{-4}$ | $10^{-3}$ | $10^{-2}$ | $10^{-1}$ |
| *Conv-2(sub-sample rate 16)* | | | | | | *Conv-2(sub-sample rate 16)* | | | | | |
| 0 | 28.43 | 30.65 | 22.09 | 14.35 | 1.05 | 0 | 94.53 | 94.40 | 96.38 | 97.32 | 99.78 |
| $10^{-4}$ | 6.71 | 11.14 | 13.17 | 9.85 | 1.32 | $10^{-4}$ | 98.66 | 97.92 | 97.64 | 98.19 | 99.71 |
| $10^{-3}$ | 6.71 | 10.92 | 10.90 | 9.73 | 1.08 | $10^{-3}$ | 98.60 | 97.87 | 98.07 | 98.20 | 99.75 |
| $10^{-2}$ | 2.45 | 4.23 | 5.61 | 4.38 | 0.66 | $10^{-2}$ | 99.48 | 99.17 | 99.91 | 98.09 | 99.82 |
| $10^{-1}$ | 0.56 | 0.87 | 0.86 | 0.85 | **0.11** | $10^{-1}$ | 99.88 | 99.82 | 99.84 | 99.82 | **99.97** |
| *Conv-2(sub-sample rate 32)* | | | | | | *Conv-2(sub-sample rate 32)* | | | | | |
| 0 | 28.61 | 18.64 | 16.70 | 3.34 | 0.52 | 0 | 94.59 | 96.49 | 96.89 | 99.32 | 99.89 |
| $10^{-4}$ | 4.60 | 8.91 | 8.89 | 3.04 | 0.43 | $10^{-4}$ | 99.08 | 98.32 | 98.34 | 99.36 | 99.89 |
| $10^{-3}$ | 4.67 | 9.29 | 9.54 | 3.43 | 0.61 | $10^{-3}$ | 99.04 | 98.19 | 98.17 | 99.27 | 99.85 |
| $10^{-2}$ | 1.62 | 3.43 | 3.42 | 1.38 | 0.24 | $10^{-2}$ | 99.66 | 99.30 | 99.28 | 98.65 | 99.92 |
| $10^{-1}$ | 0.38 | 0.76 | 0.81 | 0.37 | **0.08** | $10^{-1}$ | 99.92 | 99.84 | 99.84 | 99.91 | **99.98** |

In Table 6 we show how the FPR95 and AUROC metrics vary as a function of the different thresholds chosen for the last two convolutional layers of MobileNet-v2, for two different sub-sampling rates, 16 and 32, of the Conv-2 layer. The results reported are based on combining our HAct descriptors over the last classification (dense) layer and the two convolutional layers (Conv-1 and Conv-2), where the Conv-1 layer was sub-sampled with a rate of 32. Section 4.2.1 in the main paper has the details regarding the choice of combining these descriptors. We kept the choice of $\varepsilon = 10^{-4}$ fixed for the dense layer while we varied $\varepsilon$ for the convolutional layers. It is seen that $\varepsilon = 10^{-1}$ is the optimal choice irrespective of the sub-sampling rate for MobileNet-v2. Moreover, the sub-sampling rate of 32 on Conv-2 almost always resulted in better OOD detection performance than sub-sampling by 16, consistent with our observations in Section 4.2.1 that a lower-subsampling rate does not necessarily result in better OOD detection.

