# OpenReview forum: "HAct: Out-of-Distribution Detection with Neural Net Activation Histograms"
_ICLR.cc/2024/Conference — Submitted to ICLR 2024_

### Official Review · Reviewer_Pwoh · 2023-10-29

**Soundness:** 2 fair
**Presentation:** 3 good
**Contribution:** 3 good
**Rating:** 3
**Confidence:** 4

**Summary:**

This paper proposes a post-hoc OOD detection algorithm HAct by modeling the statistics of the activation weights of some linear layers of a pretrained network. Specifically, the class-wise average of the activation histograms is used as prototypes and are used to compare with the statistics of test images. A subsampling is employed to speed up computation. The proposed method shows impressive performance on the ImageNet-1K ID dataset using ResNet50 and MobileNet-v2.

**Strengths:**

1. The use of the Wasserstein barycenter of activation histogram as a feature to detect OOD is intuitive and the performance in Tab. 1 looks exceptionally good.
2. The presentation is easy to follow.

**Weaknesses:**

1. Some important technical details are not clear. For example,
    - (a) P4L6, "the most likely category $c^*$ ...". How is the most likely category be determined? By the softmax prediction, or by the distance between the average activatioln histograms?
    - (b) P4L12, "... to return ID if all the OOD detectors for all layers return ID". Firstly, How the $\tau$'s are determined when multiple layers are used. In the evaluation metric FPR95, the threshould should be chosen s.t. TPR=95%. Is this guaranteed when multiple $\tau$'s are used?
    - (c) How to compute AUROC when multiple layers are used? Importantly, how to aggragate the scores from multiple layers to one score? It is unclear in the text since the ID/OOD-ness from different layers are combined logically rather than numerically.
    - (d) P3 Equ. (1). How the partition values $\alpha_1, \cdots, \alpha_m$ are determined and what's the typical values of $m$ and $\alpha_i$ in experiments? What's the effect of $\alpha_i, i>1$ on performance?

2. Lack of reference and comparison to related work. The classical baselines MSP [a], Mahalanobis [b], etc. are not reviewed and compared.

3. Insufficient experiment on more network structures and on more datasets. How about models like BiT, RepVGG? Alos, the SUN and Places dataset are quite noisy as the OOD dataset for ImageNet-1K. A large portion of their images are overlapped with the ID dataset [c]. Please test on OpenImage-O [d] and ImageNet-O [e].

4. Can this method be applicable to vision transformers?

- [a] "A Baseline for Detecting Misclassified and Out-of-Distribution Examples in Neural Networks." ICLR 2016.
- [b] "A simple unified framework for detecting out-of-distribution samples and adversarial attacks." NeurIPS 2018.
- [c] "In or Out? Fixing ImageNet Out-of-Distribution Detection Evaluation". ICML 2023.
- [d] "Vim: Out-of-distribution with virtual-logit matching." CVPR 2022.
- [e] "Natural adversarial examples". CVPR 2021.

**Questions:**

Other than the questions raised in the weakness section, my main question is on the evaluation practice in the experiments. As mentioned in the weakness part, the evaluated OOD dataset is very noisy and many ID images are mixed in the OOD dataset. So it is counter-intuitive that the FPR95 could reach 0.x% and AUROC reach 99%. Please clarify the evaluation process. Specifically, (a) is the validation set of ImageNet-1K used in the computation of FPR95/AUROC? The training set for tuning the histogram should be independent of the val set used for evaluation. (b) pretend that the test set of ImageNet-1K is an OOD dataset of ImageNet-1K, and please test the AUROC of your method.

---

### Official Review · Reviewer_NrJ4 · 2023-10-31

**Soundness:** 3 good
**Presentation:** 3 good
**Contribution:** 3 good
**Rating:** 5
**Confidence:** 4

**Summary:**

This paper develops the HAct (activation histogram) descriptor to perform OOD detection according to the (Wasserstein) distribution distance between in-distribution statistics and input samples. A sub-sampling strategy is implemented to accelerate the inference process. HAct achieves a surprisingly strong OOD detection performance on the ImageNet benchmark.

**Strengths:**

1. Quantifying distributional differences of network's activations between ID and OOD samples is interesting.
2. The OOD detection performance on ImageNet benchmark (e.g., 0.03% of FPR95 and 99.98% of AUROC) is amazing.

**Weaknesses:**

1. The performance on ImageNet benchmark is extremely high, as shown in Table 1, but there is a lack of relevant discussion and explanation, which are supposed to provide valuable insights to the community.
2. Besides, as HAct performs OOD detection by comparing with the activation histogram of predicted category $c^*$ (if I did not misunderstand), now that HAct can accurately identify the samples belongs to $c^*$ or is just an OOD, the ID classification accuracy should correspondingly be extremely high, which however is not displayed in Table 1.
3. Consequently, in Table 2, HAct's performance on the relatively simpler CIFAR benchmark instead gets much worse than the more complicated ImageNet benchmark. Reasonable discussions should be given.
4. Regarding the methodology, In Section 3.1, why this paper chooses the absolute value of $|W_{ij} * x_j|$ to compute the histogram, rather than the commonly-used ReLU or the network's original activation functions? Why the bias is not considered when computing the activation histograms? If the initial network has bias parameters, will all the biases just be ignored?

**Questions:**

1. According to Section 3.1, it seems that this paper computes the activation histogram of $A_{ij}$, regardless of either input channel i or output channel j. Why not calculate finer histograms regarding i and j?

---

### Official Review · Reviewer_XeRd · 2023-11-01

**Soundness:** 2 fair
**Presentation:** 3 good
**Contribution:** 3 good
**Rating:** 3
**Confidence:** 3

**Summary:**

The paper proposes an approach for detecting samples coming from an out of domain distribution. The authors propose a descriptor HAct - activation histograms, which outperforms state of the art methods by 25% and reaches close to 100% accuracy on standard metrics used for this task. The idea is to compute barycenters of activation histograms for each class in the training dataset. During inference, we measure the distance between this histogram and the test examples and classify the sample as in-domain and out of domain.

**Strengths:**

The paper, if I am reading the results correctly, essentially claims to solve the problem all together. The method is also fairly simple, so that is also a good thing.

**Weaknesses:**

I am a bit skeptical of the results which are presented.

**Questions:**

If Table 1 is correct, then FPR95 is almost zero using this method on a dataset like Places, which means that all high confidence false positives can be detected using this algorithm. In other words, we have 0.02% false positives at 95% recall for OOD detection. If this is the case, one could do leave one out class training and obtain 95% accuracy on Places. Am I missing something here? Can we also show that we can obtain 95% accuracy on Places, which would also be state of the art on this dataset (and others?) by a large margin for both MobileNet and ResNet.

---

### Meta-Review · Area_Chair_SLie · 2023-12-05

**Metareview:**

All reviewers are negative about the paper. No rebuttal is provided. Thus, the paper is rejected.

**Justification For Why Not Higher Score:**

All reviewers are negative about the paper. No rebuttal is provided. Thus, the paper is rejected.

**Justification For Why Not Lower Score:**

N/A

---

### Decision · Program_Chairs · 2024-01-16

Reject